# RoI-MedCap: Region of Interest-Based Medical Image Captioning with Multi-Stream Connector

Al Shahriar Rubel
*Department of Informatics*
*New Jersey Institute of Technology*
Newark, NJ, USA
ar2633@njit.edu

Frank Y. Shih
*Department of Computer Science*
*New Jersey Institute of Technology*
Newark, NJ, USA
shih@njit.edu

Fadi P. Deek
*Department of Informatics*
*New Jersey Institute of Technology*
Newark, NJ, USA
fadi.deek@njit.edu

*Abstract*—Medical image captioning has gained significant attention due to the rapid advancements in Artificial Intelligence. However, existing research primarily focuses on global image captioning, lacking a mechanism for Region of Interest (RoI)-based captioning where users can specify an area and receive a caption centered on that specific region. In this paper, we propose a novel architecture with a vision encoder, a connector, and a Large Language Model (LLM) to generate captions for medical images with integrated RoI. We introduce a Multi-Stream Connector (MSC) to project visual features from a vision encoder to a representation that helps the LLM to generate captions centered on a specified region of an image indicated by a bounding box. We aim to generate captions with three aspects including the modality and structure, RoI analysis and lesion findings in RoI, and local-global relationship denoting impacts of findings in RoI to other regions. To achieve this goal, MSC incorporates three Cross Attentions focusing on three different aspects of generated captions. Our extensive experiments demonstrate that our method is well capable of generating captions highly aligned with human judgement, compared to existing related methods. The source code is available at https://github.com/alshahriarrubel/RoI-MedCap.

*Index Terms*—Medical Image Captioning, Radiology Report Generation, Vision Language Model (VLM), Large Language Model (LLM), Region of Interest (RoI), Multi-Stream Connector (MSC), Cross Attention, Artificial Intelligence

## I. Introduction

Medical imaging, particularly modalities like X-rays, MRI, CT scan, PET, dermoscopy, endoscopy, plays a vital role in modern healthcare, assisting in the diagnosis, monitoring, and treatment planning for numerous diseases. Automatic medical image captioning is the process of generating textual descriptions for medical images, which can significantly alleviate the burden on clinicians, particularly radiologists, who must deal with a large volume of medical images generated nearly uninterrupted.

Despite considerable progress in medical image captioning [1]–[3], there lacks thorough investigation on the Region of Interest (RoI) based captioning task that focuses on a particular part of an image and generates a caption centered on that area. Moreover, captions with consistent structure aid in rapid identification of task specific information. The task of RoI-based captioning has profound implications for various applications such as specialized radiology report generation [4], [5], educational tools [6], and assistive technologies [7]

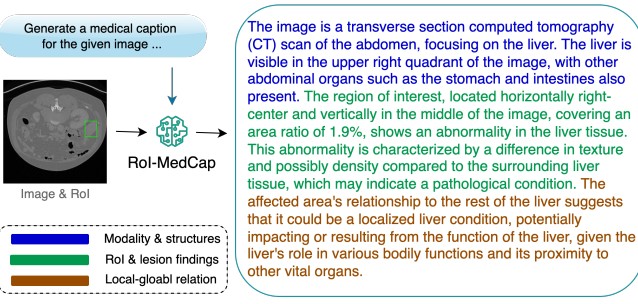

Fig. 1. Our model's capability for generating structured medical image captions, integrating modality, anatomical structures, detailed RoI analysis with lesion findings, and local-global relationships that detail the impact of RoI findings on other regions.

for enthusiastic patients and visually impaired healthcare professionals.

In this paper, we introduce an RoI-based medical image captioning approach that generates structured textual description for images with integrated RoI, as shown in Fig. 1. Our proposed architecture incorporates three key components: a Vision encoder, a Connector, and a Large Language Model (LLM). While utilizing a pre-trained vision encoder and an LLM, we introduce a novel connector specialized for the task of RoI-based structured captioning.

Our main contributions can be summarized as follows:

- We introduce a novel task of RoI-based structured captioning given medical images with integrated RoI.
- We propose a novel Multi-Stream Connector (MSC) that projects visual features from a vision encoder into word embedding space, allowing an LLM to generate specialized captions.
- We employ an efficient training strategy which involves training only the connector, without fine-tuning vision encoder and LLM.

## II. Related Works

### A. Medical image captioning

With the advent of Generative Pretrained Transformer (GPT) [8], many captioning frameworks utilize its text generation capabilities in medical image captioning tasks. A method

utilizes two captioning models, Show-Attend-Tell (SAT) and GPT-3, where SAT first identifies and focuses on critical image regions, and then GPT-3 generates a detailed report from the SAT output [9]. MSMedCap [10] utilizes a pre-trained CLIP [11] encoder to extract overall information and Segment Anything model (SAM) [12] guided encoder to capture fine-grained details. Features from CLIP and SAM are aligned and aggregated by dual Q-Formers and linear projection layers, and the OPT LLM [13] is employed to generate captions. RadTex [14], a CNN-encoder with transformer-decoder architecture with flexible prompting mechanism, explores bidirectional captioning as an alternative medical vision language pretraining strategy to contrastive learning approaches. MoColl [15] decomposes complex image captioning tasks into a series of interconnected question-answer subtasks, and introduces a collaborative framework that combines a DeepSeek [16] LLM-based agent and a specialized VQA model. The agent asks quotations to the VQA model and utilizes the answers to generate captions for given images.

### B. Vision-language models

Vision-language models (VLMs) are trained with large image-text data to accomplish multiple tasks related to image and text such as visual question answering, captioning, classification, segmentation, and image generation. With recent advancements in Large Language Models (LLMs) such as GPT-4 [17], Llama-2 [18], Vicuna [19], Qwen-LM [20], DeepSeek LLM [21], Phi-4 [22], VLMs utilize the text generation capabilities of LLMs by treating visual features as a form of language input [23]–[28]. LLaVA-Med [29] employs a curriculum learning method for adapting LLaVA [24] to the biomedical domain using biomedical multi-modal instruction-following dataset. Med-MoE [30] utilizes multiple domain-specific experts along with a global meta expert which captures global medical information to assist the specified experts. LVM-Med [31] uses a self-supervised learning approach based on second-order graph matching, while LoGra-Med [32] introduces a multi-graph alignment objective which aligns a triplet consisting of an input image, its instruction data, and its extended long-context version.

Despite comprehensive understanding and generation capabilities for multiple tasks, these models lack task and domain specific expertise, which triggers the need of specialized models for RoI-based medical image captioning. Therefore, we introduce a framework for generating structured textual descriptions of medical images with integrated RoI.

## III. THE PROPOSED METHOD

### A. Overview

The network architecture is illustrated in Fig. 2, with three key components: a Vision encoder, a Connector, and a Large Language Model (LLM). The primary goal of our work is to effectively leverage the capabilities of both the pre-trained vision encoder and LLM, and to train a connector that transforms the visual encoder's output into an effective representation as the input to the LLM. In our approach, we

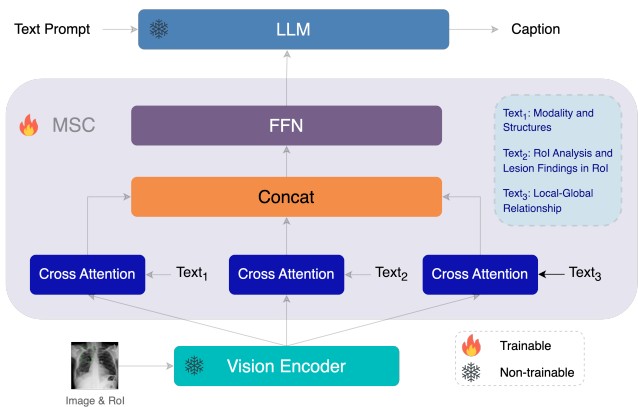

Fig. 2. Overview of our architecture with a vision encoder, a Multi-Stream Connector (MSC), and a Large Language Model (LLM) to generate structured captions for medical images with integrated RoI, using an efficient training strategy that trains only the connector.

utilize the pre-trained CLIP ViT-L/336px vision encoder [11] and LLM LLaMA3 [33] from the MedTrinity LLaVA-Medcap [34]. A Multi-Stream Connector (MSC) is employed as the connector between the vision encoder and the LLM.

For an input image, the vision encoder provides visual features and the MSC plays the role of a projector to transform the visual features into word embedding space, and reduce the modality gap between the visual and text features, connecting the vision encoder and the LLM. This enables the LLM to treat the visual features similarly to textual inputs. The cross attention modules in the Multi-Stream Connector are designated to find the visual features corresponding to various information regarding the image, specifically the modality and structures, RoI analysis and lesion findings in RoI, and local-global relationships.

### B. Multi-Stream Connector (MSC)

The Multi-Stream Connector (MSC), as illustrated in Fig. 2, contains three Cross Attention modules. Each Cross Attention module integrates image features with text embeddings. This is achieved by first employing fully connected layers to project them into a common attention space, forming Query (from text), Key, and Value (from image) representations. Subsequently, Multihead Attention (MHA) computes the relevance between the text Query and image Keys, using the attention weights to selectively combine image Values, thereby producing a contextually enriched output that fuses information from both modalities.

---

**Algorithm 1** Multihead Attention (MHA)

---
1: MHA (Q, K, V):
2: Input: Q, K, V
3: Attention: $A_i = Softmax(\frac{QW_i^Q(KW_i^K)^T}{\sqrt{d_k}})$
4: Head: $H_i = A_i V W_i^V$
5: Multi-Head: $H = [H_1, H_2, H_3, \ldots, H_h]W^H$
6: $Y = FC(H)$
7: Output: Y

---

PERFORMANCE COMPARISON WITH EXISTING METHODS. BOLD VALUES INDICATE THE BEST RESULTS AMONG COMPARED METHODS.

| Model | Parameters | Exact Match | F1 Score | Precision | Recall | BLEU-1 | BLEU-2 | BLEU-3 | BLEU-4 | Average |
|---|---|---|---|---|---|---|---|---|---|---|
| LLaVA-Med v1.5 [29] | 13B | 42.10 | 54.60 | 61.58 | 49.72 | 37.59 | 15.22 | 8.73 | 5.86 | 34.43 |
| LLaVA-Medcap [34] | 8B | 38.67 | 63. 52 | 63. 05 | 65.16 | 46.75 | 22.47 | 14.45 | 10.78 | 40.61 |
| HealthGPT-M3 [27] | 3.8B | 42.51 | 54.61 | 62.52 | 49.14 | 36.93 | 15.13 | 8.71 | 5.85 | 34.43 |
| HealthGPT-L14 [27] | 14B | 43.84 | 53.56 | 61.95 | 47.84 | 34.65 | 13.27 | 7.09 | 4.25 | 33.18 |
| HuatuoGPT-Vision-7B [28] | 7B | 34.60 | 48.85 | 45.54 | 54.74 | 36.55 | 12.12 | 6.49 | 4.24 | 30.39 |
| HuatuoGPT-Vision-34B [28] | 34B | 35.80 | 53.48 | 51.43 | 56.88 | 40.69 | 14.27 | 7.80 | 5.11 | 33.18 |
| RoI-MedCap (ours) | 8B | **44.09** | **67.89** | **69.11** | **68.00** | **50.55** | **29.29** | **21.11** | **17.04** | **45.89** |

TABLE II
PERFORMANCE COMPARISON AGAINST GEMINI WITH 3K DATA

| Model | Parameters | Exact Match | F1 Score | Precision | Recall | BLEU-1 | BLEU-2 | BLEU-3 | BLEU-4 | Average |
|---|---|---|---|---|---|---|---|---|---|---|
| Gemini Flash 2.5 [35] | Unknown | 34.30 | 50.69 | 53.77 | 49.09 | 35.83 | 12.07 | 5.94 | 3.43 | 30.64 |
| RoI-MedCap (ours) | 8B | **44.21** | **67.98** | **69.19** | **68.13** | **50.49** | **29.38** | **21.17** | **17.11** | **45.96** |

As stated in Algorithm 1, MHA takes three inputs: Query (Q), Key (K), and Value (V). Each attention head utilizes learnable projection matrices: $W_i^Q \in \mathbb{R}^{d \times d_Q}, W_i^K \in \mathbb{R}^{d \times d_K}, and\ W_i^V \in \mathbb{R}^{d \times d_V}$, where $d_Q = d_K = d_V = d/h$ and h is the number of heads. The attention mechanism involves linearly projecting Q and K, scaling their dot product by $\frac{1}{\sqrt{d_k}}$, and then applying the Softmax function to compute attention scores. These scores are subsequently multiplied by the linearly projected V. The outputs of all heads are then concatenated as shown by the [.] operation in line 5 of Algorithm 1 and linearly transformed by a final projection matrix $W^H \in \mathbb{R}^{d \times d}$. Finally, a fully connected layer (FC) transforms the output.

The first Cross Attention module takes the visual features from the vision encoder and text features from the text "modality and structures". This connector's role is to find the visual features which are relevant to the modality and structures in the given image. Similarly, the second Cross Attention module finds the visual features relevant to RoI analysis and lesion findings in the RoI. Furthermore, the third Cross Attention module is assigned to find the visual features relevant to local-global relationships, denoting the potential impact of the abnormalities in the RoI to other parts of the given image. The output features from all the Cross Attention modules are concatenated as follows:

$$F = [F_1, F_2, \ldots, F_N] \tag{1}$$

In Equation (1), $F_i$ is the output feature of the i-th cross attention and N is the number of cross attentions. The concatenated features undergo a projection via a Feed Forward Network (FFN) to align their dimension with that of the LLM inputs. As detailed in Algorithm 2, the FFN comprises fully connected (FC) layers and an activation function, $\sigma$, denoting GELU.

## IV. EXPERIMENTS AND RESULTS

### A. Implementation Details

During the training, the vision encoder and the LLM are frozen and only the connector is trained. While utilizing the

---

**Algorithm 2** Feed Forward Network (FFN)

1: FFN (X):
2: Input: $X$
3: $Y = FC(\sigma(FC(X)))$
4: Output: Y

---

pre-trained vision encoder and LLM from the specialized medical model LLaVA-Medcap, which was trained on medical multimodal data and fine-tuned using multi-granular annotations from MedTrinity, we train our MSC with a small segment of data from the MedTrinity Dataset. Since our model is based on the LLaVA-Medcap, which has been trained with medical data, we choose to train only the connector of our model for training efficiency. We employ four NVIDIA A100 GPUs for training our model for 40 epochs in around 44 hours.

### B. Datasets and Evaluation Metrics

We utilize a small segment of the MedTrinity-25M [34] dataset. MedTrinity-25M provides 25M image-caption pairs where bounding boxes, which indicate the RoI, are integrated with images. Specifically, we use 43K image-caption pairs from MedTrinity-25M, corresponding to the original SA-Med [36] dataset, containing different modalities including MRI, CT scan, X-ray, PET, Endoscopy, and Dermoscopy. We split the selected dataset to allocate 5000 samples for validation, another 5000 for testing, and the rest for training. For testing, we also incorporate a 3K-sample subset of MedTrinity-25M, which is derived from the FLARE23 [37], ULS23 [38] and BRATS24 [39] datasets.

To evaluate the methods, we employ several evaluation metrics including, Exact match, F1 score, Precision, Recall, BLEU-1, BLEU-2, BLEU-3, and BLEU-4 [40]. Additionally, we include the average scores of these metrics.

### C. Performance Comparison

To demonstrate a comparative performance of our method, we include the evaluation scores of our method and several baseline methods, as shown in Table I. We evaluate

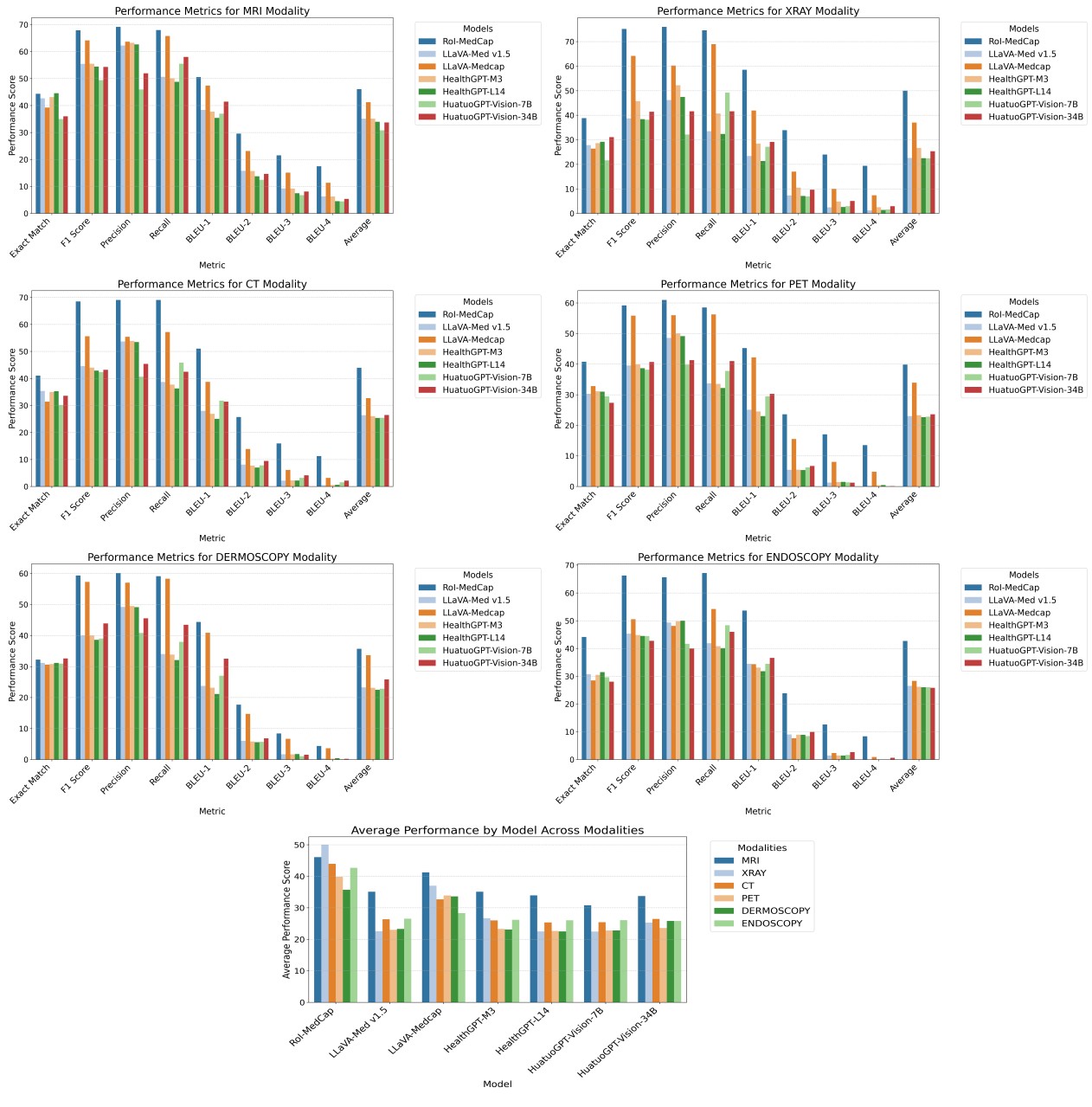

Fig. 3. Comparative performance across various imaging modalities. Higher values indicate superior performance for all metrics.

TABLE III
PERFORMANCE COMPARISON IN VARIOUS DATASETS

| Metric | FLARE23 | | ULS23 | | BRATS24 | | Mean | |
|---|---|---|---|---|---|---|---|---|
| | LLaVA-Medcap | RoI-MedCap | LLaVA-Medcap | RoI-MedCap | LLaVA-Medcap | RoI-MedCap | LLaVA-Medcap | RoI-MedCap |
| Exact Match | 33.63 | 34.81 | 29.56 | 34.69 | 34.02 | 41.13 | 32.40 | 36.88 |
| F1 Score | 57.43 | 62.13 | 53.95 | 58.77 | 62.85 | 64.06 | 58.08 | 61.65 |
| Precision | 57.89 | 62.65 | 50.12 | 55.83 | 59.70 | 64.04 | 55.90 | 60.84 |
| Recall | 58.58 | 62.89 | 59.36 | 62.95 | 67.12 | 65.03 | 61.69 | 63.62 |
| BLEU-1 | 39.85 | 42.97 | 38.78 | 43.45 | 45.72 | 48.46 | 41.45 | 44.96 |
| BLEU-2 | 15.91 | 17.18 | 13.02 | 17.35 | 18.94 | 24.89 | 15.96 | 19.81 |
| BLEU-3 | 8.73 | 8.99 | 4.31 | 9.21 | 9.64 | 15.12 | 7.56 | 11.11 |
| BLEU-4 | 5.50 | 5.33 | 1.38 | 5.17 | 5.60 | 10.78 | 4.16 | 7.09 |
| Average | 34.69 | 37.12 | 31.31 | 35.93 | 37.95 | 41.69 | 34.65 | 38.25 |

TABLE IV
ABLATION STUDY WITH VARYING NUMBER OF CONNECTORS IN RoI-MEDCAP

| No. of Connectors | Exact Match | F1 Score | Precision | Recall | BLEU-1 | BLEU-2 | BLEU-3 | BLEU-4 | Average |
|---|---|---|---|---|---|---|---|---|---|
| 1 | 43.86 | 67.71 | 69.18 | 67.48 | 50.71 | 28.98 | 20.61 | 16.39 | 45.62 |
| 2 | 45.34 | 67.14 | 69.55 | 66.08 | 50.2 | 28.6 | 20.36 | 16.26 | 45.44 |
| 3 | 44.09 | 67.89 | 69.11 | 68.00 | 50.55 | 29.29 | 21.11 | 17.04 | 45.89 |
| 4 | 43.70 | 67.61 | 69.07 | 67.44 | 50.41 | 28.66 | 20.27 | 16.11 | 45.41 |

TABLE V
ABLATION STUDY WITH VARYING NUMBER OF TRAINING EPOCHS

| No. of Epochs | Training Time (hh:mm) | Exact Match | F1 Score | Precision | Recall | BLEU-1 | BLEU-2 | BLEU-3 | BLEU-4 | Average |
|---|---|---|---|---|---|---|---|---|---|---|
| 1 | 01:17 | 42.28 | 65.28 | 67.23 | 64.77 | 47.25 | 23.98 | 15.64 | 11.41 | 42.23 |
| 10 | 09:05 | 43.20 | 67.22 | 68.49 | 67.10 | 50.57 | 28.11 | 19.69 | 15.52 | 44.99 |
| 20 | 19:56 | 43.07 | 68.05 | 68.89 | 68.53 | 50.57 | 28.93 | 20.60 | 16.50 | 45.64 |
| 30 | 30:48 | 45.57 | 67.10 | 69.50 | 66.02 | 50.28 | 28.94 | 20.78 | 16.70 | 45.61 |
| 40 | 43:38 | 44.09 | 67.89 | 69.11 | 68.00 | 50.55 | 29.29 | 21.11 | 17.04 | 45.89 |

our method against recent medical vision language models such as LLaVA-Med [29], LLaVA-Medcap [34], HealthGPT-M3 [27], HealthGPT-L14 [27], HuatuoGPT-Vision-7B [28], HuatuoGPT-Vision-34B [28]. For consistent comparison, all models are evaluated on the same test dataset and text prompt used for our method. In Table II, we also compare the performance of our method against Gemini Flash 2.5 [35] evaluating with a 3K test dataset. We can observe that our method shows significant quantitative improvement compared to the existing relevant methods. Moreover, RoI-MedCap consistently outperformed other models across most medical imaging modalities, as depicted in Fig. 3, showcasing its robust performance. Furthermore, we extend our evaluation to demonstrate better generalization across datasets, with additional unseen MedTrinity subsets derived from the FLARE23 [37], ULS23 [38] and BRATS24 [39] datasets. The results, as shown in Table III, consistently demonstrate the robust performance of our method in various datasets, outperforming the MedTrinity LLaVA-Medcap.

### D. Ablation Studies

To validate the effectiveness of the components in our architecture, we conducted a quantitative ablation study. Table IV shows that the choice of three connectors in the MSC provides best average performance. Moreover, we conducted another ablation study to investigate the performance of our method with a varying number of training epochs, as shown in Table V. It is evident that training our method for only one epoch is sufficient to achieve performance superior to the existing methods listed in Table I, highlighting the inherent advantages of our chosen architecture.

### E. Our Connector vs LLaVA-Med Connector

Our connector is more effective than the two-layer MLP used in the MedTrinity LLaVA-Medcap. Table VI presents the quantitative results, demonstrating that our connector achieves higher evaluation scores even though it was trained with fewer data than the MLP. The MLP had prior training and fine-tuning

with MedTrinity data, and for this comparison, we further fine-tuned it using the same data as our method.

### F. Qualitative Analysis

Fig. 4 presents some example captions generated by our method RoI-MedCap and other methods. We highlight the generated words based on matches and mismatches with reference captions. Green, Yellow, and Orange colors represent significant match, partial match, and significant mismatch with reference caption, respectively. While both LLaVA-Med and LLaVA-Medcap failed to identify the modality of the given computed tomography (CT) image, our RoI-MedCap correctly identifies it. Also, our method successfully detects the position and size of the RoI while other methods struggle to do the same. Furthermore, our method is able to include more comprehensive lesion findings and their impacts on other areas. Our method clearly demonstrates great potential for RoI-based structured medical image captioning.

## V. DISCUSSIONS AND LIMITATIONS

Both the quantitative results and qualitative analysis demonstrate a promising potential of our method, RoI-MedCap, in generating structured captions for medical images with integrated RoI. Compared to relevant state-of-the-art models, our method shows superior performance with successfully detecting modality and structures, position and size of the RoI, lesion findings in RoI, and local-global relationship that indicates the impacts of findings in RoI on other regions of the given images. Moreover, several ablation studies support our choice of the Multi-Stream Connector with three cross attentions, which is also more effective than the connector in LLaVA-Medcap of MedTrinity. Furthermore, our connector-only training strategy allows different levels of efficiency with preferred number of epochs, depending on the availability of resources and expected quality of generated captions.

Our method, which relies on existing training datasets and pre-trained vision and language models, has limitations for direct clinical use. For large vision language models (LVLMs) to be safe and reliable in medical applications, their evaluation

TABLE VI
CONNECTOR PERFORMANCE COMPARISON

| Connector | Model | Exact Match | F1 Score | Precision | Recall | BLEU-1 | BLEU-2 | BLEU-3 | BLEU-4 | Average |
|---|---|---|---|---|---|---|---|---|---|---|
| MLP | LLaVA-Medcap | 44.04 | 67.62 | 69.00 | 67.54 | 50.38 | 28.91 | 20.72 | 16.63 | 45.61 |
| MSC | RoI-MedCap | **44.21** | **67.98** | **69.19** | **68.13** | **50.49** | **29.38** | **21.17** | **17.11** | **45.96** |

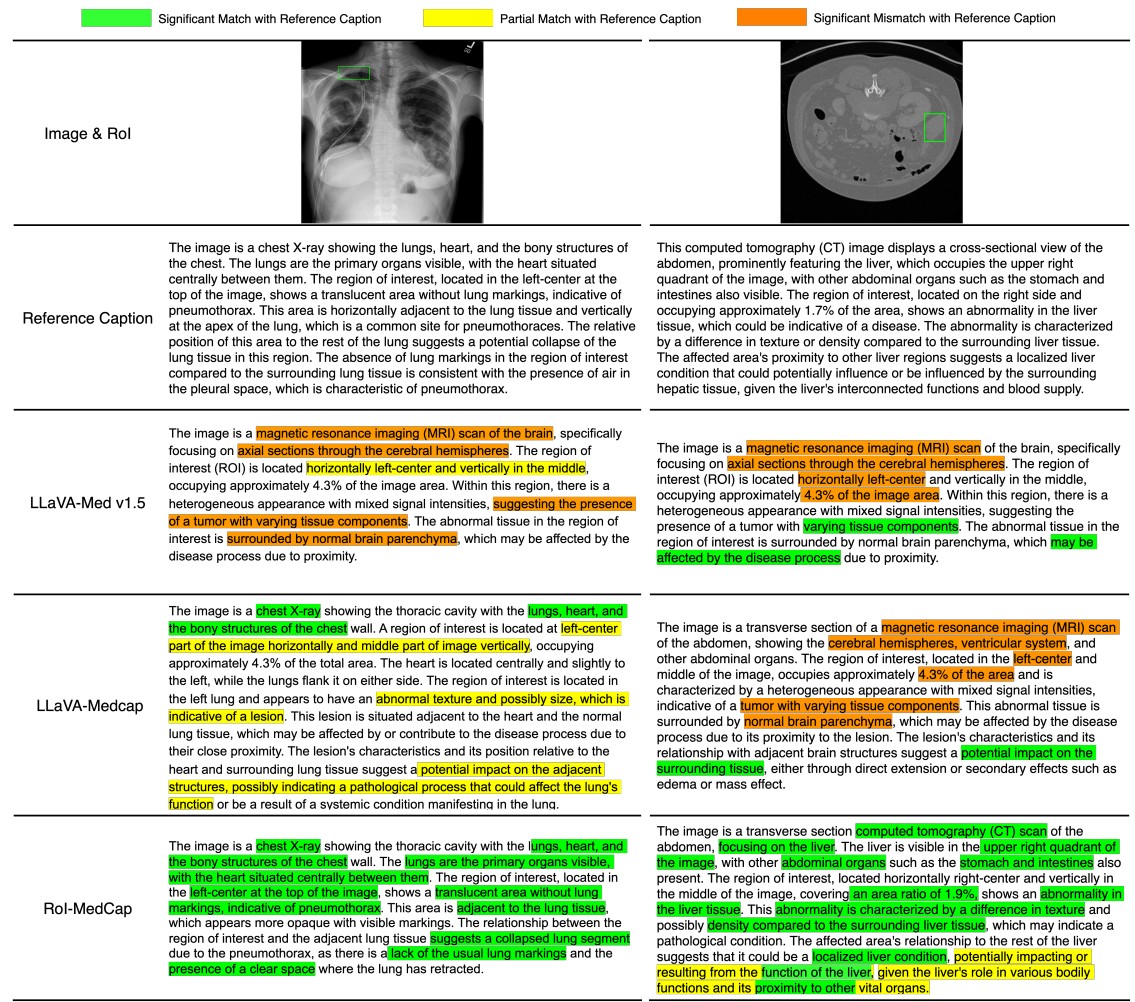

Fig. 4. Qualitative Results Comparison. Significant match, partial match and significant mismatch with reference caption are highlighted with Green, Yellow and Orange colors, respectively.

should extend beyond traditional benchmarks, with meaningful involvement of diverse healthcare professionals. Despite these challenges, our model shows promise as an assistive tool for radiologists, clinicians, and patients, serving as a foundational step for future clinical investigations.

## VI. CONCLUSION

In this paper, we have proposed a new architecture to generate captions for images with Region of Interest (RoI). We introduced the Multi-Stream Connector (MSC) that takes visual features from vision encoder and transforms them into representations that enable LLM to generate captions integrating modality and structure, RoI analysis and lesion findings in RoI, and local-global relationship. We employed an efficient training strategy in which only the connector was trained, keeping the vision encoder and the LLM frozen. From our extensive experiments, it is evident that MSC is helpful in collecting relevant information from visual features, and connecting vision encoder and LLM effectively for the task of RoI-based medical image captioning in which specific formatted captions are preferred. For future work, our aim is to scale this architecture to train models on significantly larger datasets, incorporating images from various modalities.

## ACKNOWLEDGMENT

We thank the High Performance Computing (HPC) facility at New Jersey Institute of Technology (NJIT) for providing technical support and computing resources.

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
