# OpenReview forum: "RoI-MedCap: Region of Interest-Based Medical Image Captioning with Multi-Stream Connector"
_IEEE.org/EMBS/BHI/2025/Conference — BHI 2025_

### Official Review · Reviewer_7dDt · 2025-07-17
**A Medical Image Captioning Framework via RoI-Guided Cross-Modal Alignment and LLM Integration**

**Confidence:** 5
**Clarity Of Writing:** great
**Clinical Significance:** excellent
**Methodological Novelty:** excellent
**Overall Rating:** 8

**Experiments And Results:**

excellent

**Questions For The Authors:**

1. Since both training and evaluation are conducted on the MedTrinity-25M dataset, have you tested RoI-MedCap on any external or real-world datasets to assess its generalizability?
2. How sensitive is the model to inaccurate or misaligned RoIs (if the first image caption model go wrong)? Have you evaluated performance using imperfect or noisy region annotations (e.g., simulating detection errors from automated tools)?
3. Beyond varying the number of connectors and training epochs, have you explored alternative MSC designs—such as different numbers of layers, hidden dimensions, or activation functions? Given that MSC is the only learnable component, more architectural analysis could provide valuable insights.

**Strengths:**

1. The proposed **Modality Shift Connector (MSC)** effectively bridges pretrained vision encoders and LLMs, enabling structured caption generation without the need for full model fine-tuning. This design makes the framework lightweight, adaptable, and easy to deploy.
2. The model is evaluated across a wide range of medical imaging modalities (MRI, CT, X-ray, PET, endoscopy, and dermoscopy) using the MedTrinity-25M dataset. The authors present thorough comparisons against state-of-the-art methods, demonstrating consistently competitive or superior performance.

**Summary Of The Paper:**

This paper introduces RoI-MedCap, a novel framework for structured medical image captioning that incorporates region-of-interest (RoI) features and integrates them with a large language model (LLM). Central to the framework is the Modality Shift Connector (MSC), a lightweight module that projects visual features from a pretrained vision encoder into the word embedding space of the LLM. This design enables the generation of clinically coherent, RoI-specific descriptions without requiring any fine-tuning of the vision encoder or the LLM itself. The model is trained in a parameter-efficient manner, updating only the MSC. Extensive experiments on diverse modalities (including MRI, CT, X-ray, PET, endoscopy, and dermoscopy images) sourced from the MedTrinity-25M dataset demonstrate that RoI-MedCap consistently outperforms prior medical image captioning approaches across multiple evaluation metrics (e.g., F1 score, BLEU), while maintaining a highly modular and efficient training setup.

**Weaknesses:**

1. The framework’s performance is tightly coupled with the quality and alignment of RoIs. However, the manuscript provides limited analysis on how noisy or misaligned RoIs may impact captioning accuracy. This issue is further compounded by the fact that both training and validation are conducted on the same dataset (MedTrinity-25M), raising concerns about generalizability to real-world clinical settings or external datasets.
2. Although the authors include an ablation study on the number of connectors and training epochs, they do not investigate deeper architectural variations of the MSC, such as the number of layers, activation functions, or projection mechanisms. Given that MSC is the core trainable component in the framework (the image encoder and LLM remain frozen) further exploration of its design would strengthen the technical contributions and insights of the study.

---

### Official Review · Reviewer_qG6Y · 2025-07-17
**Impressive result**

**Confidence:** 4
**Clarity Of Writing:** good
**Clinical Significance:** good
**Methodological Novelty:** good
**Overall Rating:** 7

**Experiments And Results:**

great

**Questions For The Authors:**

First, how does the MSC’s cross-attention mechanism specifically handle the projection of visual features into the LLM’s word embedding space, and what makes it more effective than the LLaVA-Med connector? Second, what steps are planned to evaluate RoI-MedCap in real-world clinical settings, particularly involving diverse healthcare professionals, as mentioned in the limitations? Third, how does RoI-MedCap handle variability across medical imaging modalities (e.g., MRI vs. dermoscopy), and were there modality-specific performance differences in the experiments?

**Strengths:**

Introducing RoI-based structured captioning is a significant contribution, addressing a gap in medical image captioning by focusing on user-specified regions, which is valuable for radiology reports, education, and assistive technologies.
Training only the MSC while keeping the vision encoder and LLM frozen reduces computational costs, achieving strong performance with just 40 epochs on four NVIDIA A100 GPUs in ~4 hours.
The MSC’s three cross-attention modules effectively capture modality/structure, RoI lesion findings, and local-global relationships, enhancing caption specificity and relevance.
Quantitative results (Tables I, II) show RoI-MedCap outperforming state-of-the-art models like LLaVA-Med and HealthGPT across multiple metrics, with ablation studies (Tables III, IV) validating the MSC design and training efficiency. Figure 4 demonstrates RoI-MedCap’s ability to correctly identify modalities (e.g., CT scans) and provide detailed RoI descriptions, surpassing baselines in precision and comprehensiveness.

**Summary Of The Paper:**

The paper proposes RoI-MedCap, a novel framework for Region of Interest (RoI)-based medical image captioning, addressing the limitation of existing methods that focus on global image descriptions. RoI-MedCap generates structured captions for specific regions of medical images (e.g., X-rays, MRIs, CT scans) by integrating a vision encoder (CLIP ViT-L/336px), a Multi-Stream Connector (MSC) with cross-attention modules, and a Large Language Model (LLM, LLaMA3 from MedTrinity LLaVA-Med). The MSC projects visual features into the LLM's word embedding space, enabling captions that describe modality/structure, RoI-specific lesion findings, and local-global relationships. The system trains only the MSC, keeping the vision encoder and LLM frozen, for efficiency. Evaluated on a subset of the MedTrinity-2SM dataset (4% of 2.5M image-caption pairs), RoI-MedCap outperforms baselines like LLaVA-Med and MedTrinity Captioner across metrics like BLEU-1 to BLEU-4. Qualitative results highlight its ability to accurately identify modalities and lesion details. Limitations include the need for real-world clinical evaluations and scalability to larger datasets.

**Weaknesses:**

Training and evaluation use only 4% of the MedTrinity-2SM dataset, raising concerns about generalizability. The paper does not discuss performance on diverse or larger datasets.
While quantitative and qualitative results are promising, the paper acknowledges the need for clinical evaluations involving healthcare professionals, which is critical for medical applications but not addressed.
The MSC’s cross-attention mechanism and feature projection process lack detailed technical explanations, making it hard to assess the novelty or complexity of the approach.

---

### Official Review · Reviewer_fSYy · 2025-07-18
**RoI-MedCap: Region of Interest-Based Medical Image Captioning with Multi-Stream Connector**

**Confidence:** 4
**Clarity Of Writing:** fair
**Clinical Significance:** good
**Methodological Novelty:** good
**Overall Rating:** 3

**Experiments And Results:**

fair

**Questions For The Authors:**

Section 4F. How many video text pairs were such qualitative comparisons performed? Were they done by a domain expert?
How does your method generalize to other datasets and imaging modalities?

**Strengths:**

- The identification of the problem is strong, the explaination for why it is important is emphasized.
- The MSC architecture is well explained and justified
- A variety of evaluation techniques are used

**Summary Of The Paper:**

The paper introduces RoI-MedCap, a novel architecture for medical image captioning that focuses on user-selected Regions of Interest (RoIs). Designed to overcome the limitations of methods that analyze the entire image, this system allows a user to specify a particular area and receive a detailed, localized description. The architecture consists of a vision encoder, a Large Language Model (LLM), and a specialized Multi-Stream Connector (MSC). The MSC acts as a bridge, utilizing three Cross Attention modules to interpret the image's modality and structure, analyze the RoI for specific findings like lesions, and understand the context of the RoI within the broader image. This process equips the LLM to generate comprehensive and highly structured captions. The paper reports that experimental evaluations show RoI-MedCap's superior performance over competing models, positioning it as a promising assistive tool for clinicians and patients.

**Weaknesses:**

1. Figure 3: The graphs are used improperly and are difficult to read. The graphs are not displaying time series data, yet the line graphs are used. Is
2. Emphasize the gold standard in your graphs and in your data.
3. Section 4E. Doesn't the data here negate the work you have done in this paper? Please explain how Table V supports your claim.

---

### Official Review · Reviewer_F3A4 · 2025-07-20
**Review of "RoI-MedCap: Region of Interest-Based Medical Image Captioning with Multi-Stream Connector"**

**Confidence:** 3
**Clarity Of Writing:** fair
**Clinical Significance:** good
**Methodological Novelty:** fair
**Overall Rating:** 5

**Experiments And Results:**

good

**Questions For The Authors:**

- In the explanation of III. THE PROPOSED METHOD, the details, internal workings, and mechanisms of the MSC appears to be overly simplified. To enhance the novelty of the proposal, could the description be strengthened to include mathematical algorithms and parameter requirements, etc?
- It appears that this paper does not delve into a comparative analysis of other potential datasets that were considered, nor does it explain why MedTrinity-25M outperformed other datasets beyond RoI integration and modality diversity. Could these explanations be strengthened?

**Strengths:**

- A new challenge is proposed with the introduction of a new task of RoI-based structured captioning for medical images integrated with RoI.
- ROI-MedCap utilizes existing pre-trained components such as vision encoders and LLMs, but it proposes a method that is superior to a simple MLP connector by framing the structured output requirements (modality, RoI findings, and local-global relationships) within the problem setting. The evaluation results demonstrate that it consistently outperforms other state-of-the-art medical visual language models, including those that can be considered more direct combinations of vision encoders and LLMs.
- An efficient training strategy is adopted, where only the MSC is trained while keeping the pre-trained vision encoder and LLM frozen, effectively proposing a strategy for training efficiency.

**Summary Of The Paper:**

This paper presents a proposal regarding medical captioning. Specifically, while existing research primarily focuses on global image captioning, often lacking the functionality for users to specify a Region of Interest (RoI) and receive captions centered on that specific area, this paper introduces a novel architecture called ROI-MedCap for generating structured medical image captions integrated with RoI. Not only does it introduce a new task of RoI-based structured captioning for medical images with integrated RoI, but it also serves as a novel core by bridging the gap between the visual features of the vision encoder and the text generation capabilities of the Large Language Model (LLM) through the Multi-Stream Connector (MSC).  To evaluate the generalization performance across different types of medical images, the evaluation dataset (the Medtrinity-25M dataset) includes various medical imaging modalities such as MRI, CT scans, X-rays, PET scans, endoscopy, and dermoscopy. The results demonstrate that ROI-MedCap consistently outperforms other models across most medical imaging modalities, showcasing the superiority of the proposed method.

**Weaknesses:**

- The Multi-Stream Connector (MSC) is introduced as a core and novel component of the ROI-MedCap architecture; however, the explanation of its internal workings appears to be overly simplified, making the novelty of the method less clear.
- The model demonstrates robust performance across various modalities within the Medtrinity-25M dataset, comparing it with other models on specific test sets; however, there is no explicit presentation or discussion regarding the formal evaluation of generalization across datasets.
- While the potential data bias in the current evaluation and challenges in practical implementation are briefly mentioned, the limitations of the evaluation dataset (Medtrinity-25M) are not sufficiently discussed, and the description regarding the breakdown of potential challenges is weak.